# Schistosoma mansoni infection induces plasmablast and plasma cell death in the bone marrow and accelerates the decline of host vaccine responses

Fungai Musaigwa[1,2,3], Severin Donald Kamdem[1,2,3,4,5], Thabo Mpotje[1,2,3], Paballo Mosala[1,2,3ᴼ], Nada Abdel Aziz[1,2,3,6ᴼ], De'Broski R. Herbert[7], Frank Brombacher[1,2,3,5], Justin Komguep Nono[1,2,8]*

1 Division of Immunology, Health Science Faculty, University of Cape Town, Cape Town, South Africa, 2 Immunology of Infectious Diseases Unit, South African Medical Research Centre, Cape Town, South Africa, 3 Cape Town Component, International Centre for Genetic Engineering and Biotechnology, Cape Town, South Africa, 4 Division of Microbiology and Immunology, Department of Pathology, University of Utah School of Medicine, Salt Lake City, Utah, United States of America, 5 Wellcome Centre for Infectious Diseases Research in Africa and Institute of Infectious Disease and Molecular Medicine, Faculty of Health Sciences, University of Cape Town, Cape Town, South Africa, 6 Chemistry Department, Faculty of Science, Cairo University, Giza, Egypt, 7 Department of Pathobiology, School of Veterinary Medicine, University of Pennsylvania, Philadelphia, Pennsylvania, United States of America, 8 Laboratory of ImmunoBiology and Helminth Infections, Institute of Medical Research and Medicinal Plant Studies, Ministry of Scientific Research and Innovation, Yaoundé, Cameroon

ᴼ These authors contributed equally to this work.
* Justkoms@yahoo.fr

**Data Availability Statement:** Data (identity and interview recorded) cannot be shared publicly because of ethical requirements of participant

## Abstract

Schistosomiasis is a potentially lethal parasitic disease that profoundly impacts systemic immune function in chronically infected hosts through mechanisms that remain unknown. Given the immunoregulatory dysregulation experienced in infected individuals, this study examined the impact of chronic schistosomiasis on the sustainability of vaccine-induced immunity in both children living in endemic areas and experimental infections in mice. Data show that chronic Schistosoma mansoni infection impaired the persistence of vaccine specific antibody responses in poliovirus-vaccinated humans and mice. Mechanistically, schistosomiasis primarily fostered plasmablast and plasma cell death in the bone marrow and removal of parasites following praziquantel treatment reversed the observed cell death and partially restored vaccine-induced memory responses associated with increased serum anti-polio antibody responses. Our findings strongly suggest a previously unrecognized mechanism to explain how chronic schistosomiasis interferes with an otherwise effective vaccine regimen and further advocates for therapeutic intervention strategies that reduce schistosomiasis burden in endemic areas prior to vaccination.

## Author summary

Schistosoma mansoni (S. mansoni), a schistosomiasis disease-causing parasite species, is most common in sub-Saharan Africa. Schistosoma mansoni can influence immune

anonymity. Anonymous interview data for all participants are available upon request to the National Ethics Committee for Human Health Research of Cameroon (http://cdnss.minsante.cm/?q=fr/content/procedure-dobtention-dune-clairance-ethique) using the contact email cnethique_minsante@yahoo.fr. All other relevant data are within the manuscript and its supporting information files.

**Funding:** This project is part of the EDCTP2 program supported by the European Union, through grant number TMA2016CDF-1571, with support from grant FLR\R1\191058 of the FLAIR Fellowship Program, a partnership between the African Academy of Sciences and the Royal Society funded by the UK Government's Global Challenges Research Fund, and the Poliomyelitis research Foundation of South Africa (Grant Nr 18/19) to JKN. FM is a recipient of a Ph.D. fellowship from the South African National Research Foundation (NRF) and funding assistance from the Univer-sity of Cape Town (UCT) and the Poliomyelitis Research Foundation of South Africa (PRF). SDK is a former recipient of a PhD fellowship from the International Centre for Genetic Engineering and Biotechnology (ICGEB), Cape Town component, and was a fellow of the Royal Society of Tropical Medicine and Hygiene (RSTMH) small grant supported by the National Institute for Health Research (NIHR) grant number (RSTMH\2019\12837708) and a Postdoctoral fellow within a UK Royal So-ciety FLAIR-funded grant (FLR\R1\191058) at the time of the present work. FB is funded by the International Centre for Genetic Engineering and Biotechnology, Cape Town component; the South African National Research Foundation and Medical Research Council with further support from CIDRI-Africa (grant No 203135/Z/16/Z). The funders had no role in study design, data collection and analysis, decision to publish, or preparation of the manuscript.

**Competing interests:** I have read the journal's policy and the authors of this manuscript have the following competing interests: Nono JK is a founding member of JRJ Health, a health-promoting association based in Cameroon. JRJ had no role in the conceptualization, design, analysis of collected data, decision to publish, and preparation of the manuscript. All the remaining authors declare no financial or non-financial competing interests.

responses and trigger physiological imbalances in their human and animal hosts, which improve their survival and multiplication in the host. These influences can disrupt the host's ability to maintain long term protective immunity mounted by vaccines for infectious diseases. Here, we investigated the impact of *S. mansoni* infection on poliovirus vaccine immunity in school-aged children and mice. We found that the parasite reduced its host's ability to maintain protective blood antibodies produced by immune responses to poliovirus vaccines. We also found that *S. mansoni* infection reduces the maintenance of antibody-producing plasma cells in the bone marrow of vaccinated mice. Our data showed that treating *S. mansoni* infected children and mice with praziquantel mitigated the parasite's negative influences on vaccine immunity. These findings suggest that in regions where schistosomiasis is endemic, the *Schistosoma* spp. parasites may be notable causes of suboptimal viral vaccine immunity maintenance by children, leaving them vulnerable to vaccine-preventable diseases.

## Introduction

Vaccines prevent highly infectious and fatal diseases which are potentially resource-demanding in the event of uncontrolled outbreaks [1]. The critical impact of vaccination on improving public health has been hailed as second only to clean water supplies [2]. With an estimated 2–3 million lives saved annually through vaccination [3], vaccines are integral disease prevention tools, particularly in developing resource-limited countries [4].

Supported by the World Health Organization (WHO), the Expanded Programme on Immunization (EPI) has been instrumental in the prevention and control of infectious diseases globally [5]. However, over 1.5 million children from developing countries die from vaccine preventable diseases annually [3]. Increasingly, clinical evidence highlights lower responses to vaccines in children from developing countries when compared to those from developed countries [4,6,7]. Among the many associated factors, parasitic helminths have been shown to jeopardize the protective memory immunity mounted by several vaccines in children [8–12].

Concerningly, the reliance on EPI vaccines in developing countries often overlaps with the endemicity for helminths such as *Schistosoma* spp parasites [13]. Over 90% of all schistosomiasis disease cases, a highly debilitating disease with an estimated 258 million people infected globally [14], occur in sub-Saharan Africa alone [15]. Children in developing countries are particularly at risk of contracting this fresh water-borne parasite [14–16]. Considering the suggested impact of schistosomiasis on vaccine effectiveness [8–13], the overlapping endemicity of schistosomiasis and the critical need for vaccinating people could be an impending challenge, one that could be catastrophic without strategic interventions.

Praziquantel (PZQ) chemotherapy is currently the only safe and effective treatment intervention recommended against all species of schistosomiasis infections [17,18]. Efforts to control schistosomiasis infection in endemic regions are commonly centred around mass drug administration (MDA) campaigns, under which preventive PZQ treatment is administered periodically to population groups at risk of schistosomiasis infections [19].

In this present study, we assessed the influence of *S. mansoni* infection on long-term protective immunity to the anti-poliovirus vaccine, used here as a proxy for antibody-dependent vaccination strategies. We conducted an observational clinical study in school children from an *S. mansoni* endemic area of Cameroon. We also conducted an experimental study using laboratory mice to uncover the immunological interactions between the vaccine-elicited responses and *S. mansoni* infection.

Summed up, the goal of this study was to integrate our findings into the current global anti-viral vaccination strategy, with the foresight to improve the effectiveness of antiviral vaccination programs, particularly in developing countries where schistosomiasis is endemic.

## Results

### Infection with *S. mansoni* impairs poliovirus specific serological memory in poliovirus vaccinated school children from a schistosomiasis endemic area

Schistosomiasis infected children were previously shown to possess lower serum antibody titres following vaccination against the measles virus [8]. Therefore, we conducted this study to investigate whether individuals harbouring *S. mansoni* infection had altered poliovirus vaccine specific serological memory responses. Children residing in Bokito, an *S. mansoni* endemic rural region of Cameroon were recruited from Yoro 1 (Y1), a public school located near *S. mansoni* infested rivers in the Bokito area in 2016.

Interviewer-administered questionnaires were utilized to collect information from informed and consenting school children assisted by parents or legal guardians. Poliovirus vaccination status was confirmed by consultation of vaccination cards and recall from parents and/or guardians. Stool samples were collected and assessed for excreted *S. mansoni* eggs using the Kato Katz technique. Blood samples were collected from all children for downstream analyses. Exclusion criteria from this study included children infected with Hepatitis B or C (using rapid diagnostic tests), Malaria (using blood smear and microscopy), or co-infection with geohelminths in addition to *S. mansoni* (using Kato-Katz) and any children with missing infection status and/or interview related data.

A total of 64 children (age range: 9.2–12.7 years old) were recruited and had their blood samples collected for this first pilot study in 2016. The prevalence of *S. mansoni* among these children was found to be 82.8% (53/64). Their blood plasma anti-poliovirus specific IgG antibody titres were quantified by ELISA. We noted significantly reduced anti-poliovirus specific IgG antibody titres in *S. mansoni*-infected children when compared to non-infected children (p<0.01) (Fig 1A).

To further confirm the robustness of this observation, a subsequent study encompassing a larger sample size with children from five schools located in the same *S. mansoni* endemic region of Bokito was conducted. Similar consent processes and exclusion criteria as described for the 2016 study were employed. From a total of 336 consenting school children, 189 children were finally recruited (S1A Fig). The children in this cohort ranged from 7 to 16 years of age, with a modal age range of 10–14 years old per school (S1 Table). Thirty-three children were recruited from the 2016 study site, Y1, and the rest from surrounding schools i.e., 41 children from Yoro 2 (Y2), 48 children from Bongando (BG), 32 from Kedia (KD) and 35 children from Ediolomo (ED). The prevalence of *S. mansoni* infection was 24.2% (8/33) at Y1, 53.7% (22/41) at Y2, 6.25% (3/48) at BG, 6.25% (2/32) at KD and 2.86% (1/35) at ED (S1 Table). Measurements of blood plasma anti-poliovirus specific IgG antibody titres for this new cohort of children further revealed that *S. mansoni* was significantly associated with lowered anti-poliovirus specific IgG in older children (>14 years) (p < 0.05) (Fig 1B).

Collectively, our observations suggested that chronic *S. mansoni* infection may negatively impact serological memory immunity elicited by poliovirus vaccination in school-aged children from endemic sites.

### Anthelminthic PZQ treatment partially restores serological memory in anti-poliovirus vaccinated school children with *S. mansoni* infections

The above associative studies could not distinguish between ongoing suppression of antigen-specific antibody responses due to persistent infection versus a permanent loss of protective

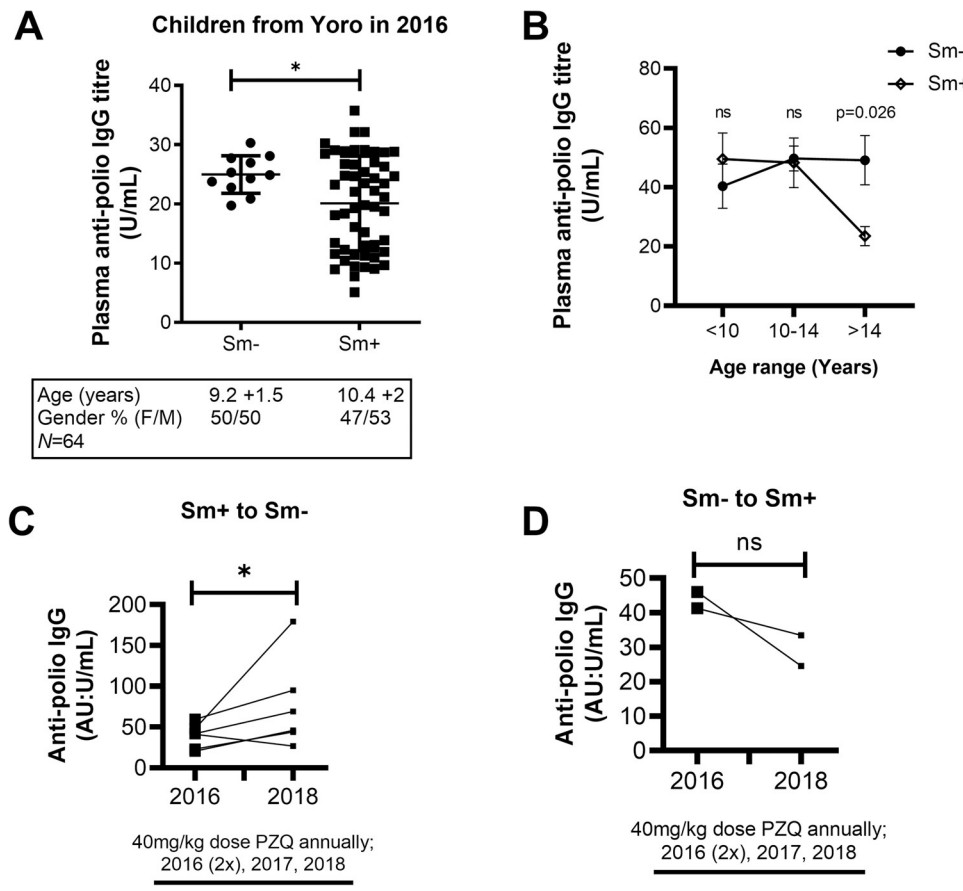

**Fig 1. Influence of S. mansoni infection and age on vaccine-induced anti-poliovirus responses in school children from five schools in a rural region of Cameroon.** (A) Anti-poliovirus IgG levels of a pilot group of school children in the village of Yoro, Bokito in Cameroon in the year 2016 (n = 64). A table indicating the population characteristics is shown below the graph. (B) Anti-poliovirus IgG titre distribution by *S. mansoni* prevalence and age groups in all children from the five additionally surveyed schools (n = 189). (C) Effect of Schistosomiasis infection, and its treatment with PZQ, on vaccine-induced plasma anti-poliovirus IgG titres in children going from egg-positive to egg-negative for *S. mansoni* over the 3 years (2016–2018) period (n = 6). (D). Effect of Schistosomiasis infection, and its treatment with PZQ, on vaccine-induced plasma anti-poliovirus IgG titres in children going from egg-negative to egg-positive for *S. mansoni* over the 3 years (2016–2018) period (n = 2). Data are expressed as mean ± S.D; All data were analysed by Shapiro-Wilk test followed by the two-tailed unpaired Student t test for unequal variances (A, B) and the two-tailed Wilcoxon signed-rank test or paired t test (C,D); ns, p> 0.05; * p< 0.05, ** p< 0.001, *** p< 0.0001; Sm, *S. mansoni*; PZQ, Praziquantel.

antibody titres. To distinguish between these possibilities, we addressed whether PZQ treatment could restore, even partially, the poliovirus vaccine specific serological memory of *S. mansoni* infected children. Praziquantel is the leading therapeutic agent for the treatment of schistosomiasis, particularly in endemic areas [20]. We conducted a follow-up investigation of children who participated in both studies performed in the years 2016 and 2018 to directly assess whether anti-polio vaccine titres would rebound post-treatment. A total of eight children with all information were identified (S1B Fig).

Six of the eight children initially found infected with *S. mansoni* in the year 2016 had cleared the infection upon follow up in the year 2018. Two of the eight children who were previously uninfected in the year 2016 were found infected with *S. mansoni* in the year 2018. Between the years 2016 and 2018, a total of 3 years had elapsed, thus translating to 3 cycles of single annual doses of PZQ (40mg/kg) treatment cycles under the MDA campaign aimed to

eliminate schistosomiasis in endemic areas of Cameroon. In addition, considering that all children in our studies were treated with single doses of PZQ (40mg/kg) outside of the MDA campaign, both S. *mansoni* infected and non-infected children effectively received an additional PZQ dose making it 4 treatment cycles over the 3-year study period.

Blood plasma anti-poliovirus specific IgG antibody titres of the selected six previously infected and now cured children were comparatively evaluated before and after treatment in relation to their parasitological status changes between 2016 and 2018 (assessed by Kato Katz in 2016 and at the end of the follow up period in 2018). We noted that these children who were previously infected with S. *mansoni* in 2016 had significantly higher anti-poliovirus specific IgG antibody titres in 2018 after having cleared the parasite (p<0.05) (Fig 1C). For the remaining two children, we noted that these children who were previously egg-negative for S. *mansoni* in 2016 had consistently lower anti-poliovirus specific IgG antibody titres in 2018 after having acquired the parasite (Fig 1D). Overall, these data suggested a positive association between PZQ treatment's S. *mansoni* worm killing and infection clearance with the partial restoration of poliovirus vaccine induced antibody titres in children who efficiently cleared their worms post treatment.

## Establishing a hexavalent (DTPa-hepB-IPV-Hib) vaccination mouse model

Given the ethical limitations of human research studies involving invasive mechanistic experimentation, we next turned to an experimental mouse model of S. *mansoni* infection to further investigate whether chronic S. *mansoni* infection actively suppressed anti-polio antibody responses. Similar to the oral poliovirus vaccine (OPV), inoculation of mice with the inactivated poliovirus vaccine (IPV) has been shown to induce robust serum anti-poliovirus IgG serological responses that can be maintained for up to 400 days [21]. We established an IPV mouse vaccination model using age and gender-matched BALB/c mice that were subcutaneously inoculated twice with a commercial hexavalent (DTPa-hepB-IPV-Hib) vaccine (Hexaxim, Sanofi Pasteur, Lyon, France) or mock treatment (1x PBS) at day 0 and day 30 (S2A Fig). The mice were bled every two weeks over 16 weeks, and their serum was assessed for anti-poliovirus specific IgG antibody titres by ELISA. Data showed significantly elevated anti-poliovirus IgG antibody titres that were maintained over the 16-week study period post-vaccination in vaccinated mice when compared to non-vaccinated naïve mice (P<0.0001) (S2B Fig). Moreover, assessment of splenic IgG1 memory B cells (S2C Fig) revealed a significant elevation of these cells as a result of vaccination, up to 16 weeks after administration of the final boost (p<0.05) (S2D Fig). This indicated that vaccination of BALB/c mice with the hexavalent (DTPa-hepB-IPV-Hib) vaccine induces long-term poliovirus specific serological memory immunity in our setting.

## Chronic S. *mansoni* infection reduces poliovirus specific antibody titres in vaccinated mice that are partially reversed by PZQ treatment

We next assessed, in mice, the impact of chronic S. *mansoni* infection and its subsequent clearance by PZQ treatment on immune responses induced by the hexavalent (DTPa-hepB-IPV-Hib) vaccine. To this end, four groups of age and gender matched BALB/c mice were established (S3 Fig). The first group represented the non-vaccinated naïve mice. The remaining three mice groups were subcutaneously injected twice with the hexavalent vaccine (DTPa-hepB-IPV-Hib) at day 0 and day 30, as per our established vaccination model (Fig 2A and 2B). One of the three groups was designated the vaccinated control (Vac+). The remaining two groups of vaccinated mice were subsequently percutaneously exposed to low dose infection with 35 S. *mansoni* cercariae at 5 days post booster vaccine administration to establish a chronic S. *mansoni* infection (Vac+Sm+).

Analysis of weight changes over the course of infection highlighted a decline from 8 weeks post-*S. mansoni*-infection (p.i), for the two groups of *S. mansoni* infected mice (Fig 2A). One cohort of Vac+Sm+ mice was administered 400mg/kg of PZQ between week 9 and 10 p.i (Vac +Sm+PZQ+), that is, after the establishment of the chronic phase of the schistosomiasis infection [22]. Upon PZQ treatment, Vac+Sm+PZQ+ mice recovered their weight within 1 week while Vac+Sm+ continued to decline until the experimental endpoint at week 18 p.i.

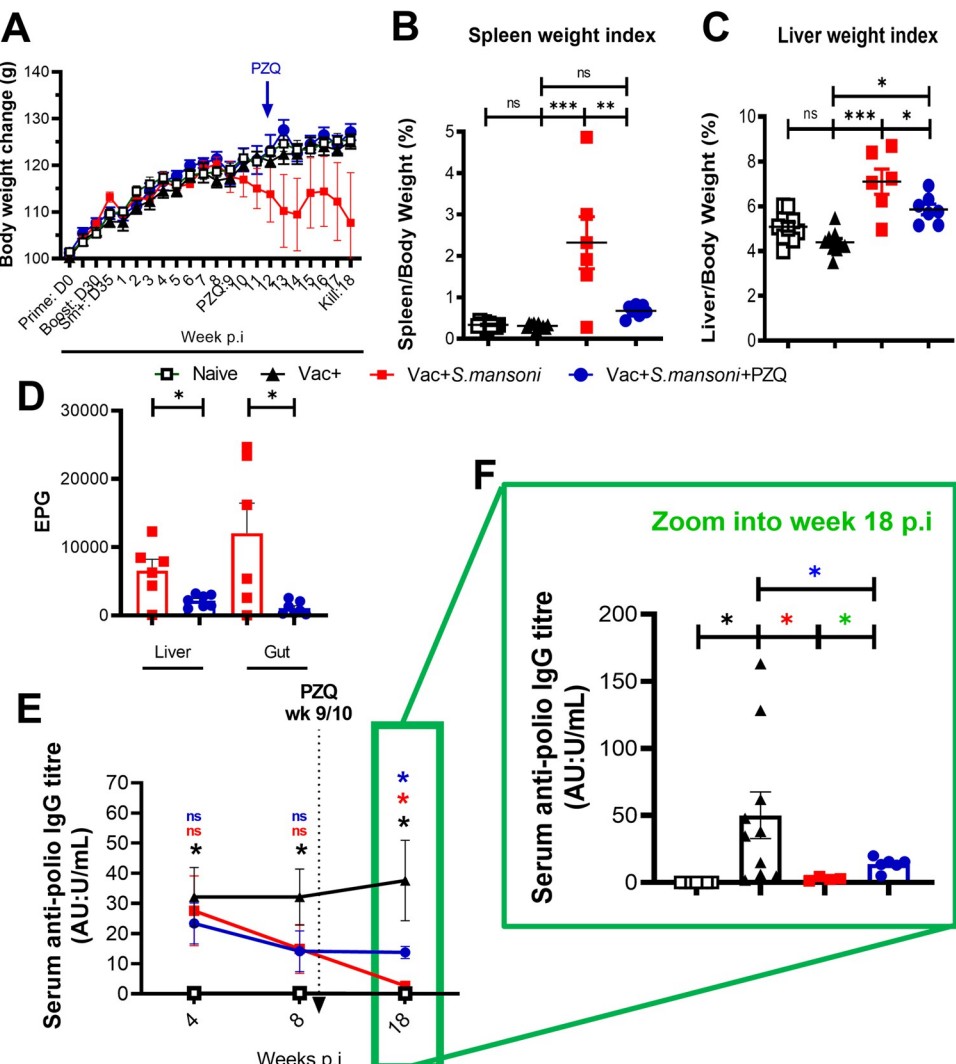

**Fig 2. Body weights, organ weights and histopathological profile of chronic *S. mansoni* infected mice post Inactivated Poliovirus Vaccine (IPV) immunisation and after treatment with PZQ.** (A) Body weight changes over time. Note a drastic weight drop in the Vac+ Sm+ group that is rescued in the Vac+Sm+PZQ+ group. (B) Spleen/body weight index at week 18 p.i (C) Liver/body weight index at week 18 p.i (D) Number of eggs per gram of liver or gut at 18 week p.i (E) Serum anti-polio IgG titre kinetics response to poliovirus vaccination in schistosomiasis-infected mice for all mice at week 4, 8 and 18 p.i (F) Anti-poliovirus IgG titres at week 18 p.i Data are expressed as mean ± S.E.M and representative of 2 independent experiments (n = 4–10 mice per group in each experiment); Data was analysed by Shapiro-Wilk test. Data was analysed by one way ANOVA followed by the Bonferroni's multiple comparisons test (B-C); two tailed Student's t test for unequal variances (D); two-way ANOVA followed by FDR corrected multiple comparisons with Vac+ group as a statistical reference with (E-F) and colour codes of black (comparison Vac+ *vs* Naive), red (comparison vac+ *vs* vac+Sm+), blue (comparison vac+ *vs* vac+sm+PZQ+) and green (vac+Sm+ *vs* Vac +Sm+PZQ+); ns, p> 0.05; * p< 0.05, ** p< 0.001, *** p< 0.0001; Vac, hexavalent (DTPa-hepB-IPV-Hib) vaccine; Vac +, vaccinated; Sm, *S mansoni*; Sm+, infected with *S. mansoni;* PZQ, praziquantel; PZQ+, treated with PZQ;p.i, post *S. mansoni* infection.

At week 18 p.i, Vac+Sm+PZQ+ mice had significantly lower spleen and liver weights than Vac+Sm+ mice ($p<0.001$ and $p<0.05$, respectively) (Fig 2B and 2C). Additionally, Vac+Sm + mice had significantly increased spleen and liver weights than Vac+ mice ($p<0.0001$), which on the other hand were within normal limits as demonstrated by comparable spleen and liver weights of naïve mice. Further, there were no notable differences in the spleen weights of Vac +Sm+PZQ+ and Vac+ mice ($p>0.05$) Fig 2B) while a minimally significant difference was observed for liver weights between the same groups of mice ($p<0.05$) Fig 2C). Chronic schisto-somiasis was confirmed by the recovery of *S. mansoni* eggs in the gut and the liver of infected mice. By week 18 p.i, PZQ treated mice showed significantly reduced *S. mansoni* egg burden in the liver and the intestines when compared to non-treated mice, thus demonstrating the efficiency of our PZQ treatment regimen (Fig 2D).

Next, we investigated the relationship between chronic *S. mansoni* infection and the para-sites' clearance by PZQ treatment on the levels of anti-poliovirus vaccine induced serological responses. Blood serum was obtained from all mice cohorts at weeks 4, 8, and 18 p.i, and serum ELISAs were performed to quantify anti-poliovirus specific IgG antibody titres over the 18 week infection period. The successful vaccination of the mice was confirmed by observed significantly elevated anti-poliovirus specific IgG antibody titres of Vac+ in comparison to naïve (non-vaccinated) mice at week 4, 8 and 18 p.i ($p<0.01$) (Fig 2E). Additional comparisons among the vaccinated groups revealed a significant decline in anti-poliovirus specific IgG anti-body titres over time in *S. mansoni* infected mice (Vac+Sm+ and Vac+Sm+PZQ+) (Fig 2E). Strikingly, by week 18 after *S. mansoni* infection, anti-poliovirus specific IgG antibody titres in both vehicle treated and PZQ-treated infected mice were significantly lower than that of Vac + mice ($p<0.01$ and $p<0.05$, respectively).On the other hand, PZQ treated Vac+Sm+PZQ + mice had significantly higher titres than their non-treated Vac+Sm+ counterparts at week 18 p.i ($p<0.05$; Fig 2E and 2F). Furthermore, associative analyses between serum anti-parasite levels (directed against Schistosoma Egg Antigen, SEA) and vaccine-elicited anti-polio IgG antibodies showed a progression of a positive association that gradually changed into a nega-tive association over time (from week 4, week 8 to week 18 p.i, S4A–S4C Fig). This could be ascribed to egg deposition as we observed a negative association between anti-polio IgG titers and egg burden in livers (S4D Fig) and gut (S4E Fig) of infected vaccinated animals. These associations were further revealed by the plotting of SEA-specific IgG over time in our differ-ent groups of mice (S4F–S4I Fig) showing an elevation of anti-SEA IgG titers early on in infected animals that rapidly rose until week 18 p.i in Vac+ Sm+ animals but less robustly after week 8 in PZQ treated Vac+Sm+ animals, as a result of deworming (S4I Fig). Taken together, these results demonstrated that chronic *S. mansoni* infection reduces anti-poliovirus specific IgG antibodies previously induced by vaccination with the hexavalent (DTPa-hepB-IPV-Hib) vaccine in BALB/c mice. Notably, PZQ treatment significantly, though incompletely, restored vaccine-induced IgG antibody titres by week 8 post administration (week 18 p.i) aligning with its ability to reduce parasite burden in the animal tissues (Fig 2D).

## Chronic *S. mansoni* infection reduces bone marrow CD138+ plasma B cell responses in mice, and anthelminthic PZQ treatment reverses this effect

The reversible inhibition of anti-poliovirus antibody titres in *S. mansoni* infected mice prompted us to evaluate cellular immune components that could potentially provide a mecha-nistic explanation to our observations. Given that long-term vaccine specific serological mem-ory immunity is maintained by B cells such as antibody producing CD138+ plasma B cells [23–26] that niche in the bone marrow, we investigated the influence of chronic *S. mansoni* infection and PZQ treatment on specific B cell subsets. First, to unambiguously assess the

influence of *S. mansoni* infection on B cell subsets in the course of an infection, BALB/c mice were infected with a low dose chronic-driving amount of cercariae (35 cercariae each percutaneously), and bone marrow and spleen were collected at weeks 4 and 10 p.i for B cell analyses using flow cytometry. We observed that *S. mansoni* infected mice had depleted CD19+B220 + total B cells, CD138+ plasmablasts and plasma cells in their bone marrow at week 10 but not at week 4 p.i (S5 Fig).

In vaccinated mice, we noted that B cell frequencies and absolute counts were significantly increased in the bone marrow of Vac+ mice when compared to naïve controls at week 18 p.i (Fig 3A and 3B). This elevation was abrogated in Vac+Sm+ mice where B cell reduction was associated with *S. mansoni* infection. In Vac+Sm+PZQ+ mice, previously reduced B cell levels were restored to levels similar to those of Vac+ mice (Fig 3B). Strikingly, the spleen B cell compartment, although stimulated by vaccination (Vac+, p<0.1) was not as robustly affected as in the bone marrow following *S. mansoni* infection (Fig 3C). Upon quantification of CD138 + plasma B cell populations (here encompassing plasmablasts and plasma cells) by flow cytometry in bone marrow and spleen (Fig 3A), the bone marrow of Vac+ mice showed significantly higher frequencies and absolute numbers of plasma B cells as defined by Live Lymphocyte +CD19+B220+IgD-IgG1-CD138+ staining (Fig 3A) when compared to those of naïve mice (p<0.001) and Vac+Sm+ mice (p<0.05) (Fig 3D). In contrast, there were no notable differences between the frequencies of bone marrow plasma B cells of Vac+ and Vac+Sm+PZQ + mice. The latter presented with significantly higher frequencies and absolute numbers of bone marrow plasma B cells when compared to those of Vac+Sm+ mice (p<0.05) (Fig 3D). Moreover, a comprehensive screening of other B cell subsets in these groups (S6 Fig) did confirm the drastic reduction of B cells in the bone marrow of Vac+ Sm+ mice (S6B Fig) and the mild reduction (frequencies only) of these B cells in the spleen of Vac+ Sm+ mice (S6C Fig) when compared to B cells in bone marrows and spleens of Vac+ mice. However, Vac+Sm + mice showed elevated levels of IgG1 memory B cells in the spleen when compared to non-infected Vac+ mice (S6D Fig) arguing for the strong induction of humoral response by schistosomiasis. Whereas follicular B cells in the spleen of vac+ mice did expand when compared to levels in the spleen of naïve mice (S6E Fig), we failed to see the massive reduction of this B cell subset as a result of Sm infection. Marginal zone B cells in the spleen of these animals were rather solicited to expand following vaccination in the Vac+ group to be considerably depleted following *S. mansoni* infection in the Vac+Sm+ group and restored to Vac+ levels after PZQ treatment in the Vac+Sm+PZQ+ group (S6F Fig). Our observations, altogether, preferentially indicated a parasite-mediated reduction of recruitment or maintenance of plasma B cells in the bone marrow of vaccinated mice as worthy of further consideration to explain, at the B cell level, reduced polio vaccine-elicited serological IgG antibody production in *S. mansoni* infected hosts. In fact, since this effect was ameliorated by PZQ-mediated parasite killing, more support was hereby lent to our serological observations that treatment against the parasite may improve vaccine response in schistosomiasis-diseased hosts. Focusing on plasma cells and to separate the possibility of the defective migration of these plasma cells to the bone marrow from impaired survival within the bone marrow niche, we analysed a C-type Lectin CD93 that has been demonstrated to be critical in the long-term survival of antibody producing CD138+ plasma B cells within the bone marrow niche but not their migration. [23,27]. To do so, we assessed the expression of CD93 on bone marrow plasma B cells for all mice groups. We noted a reduced surface expression of CD93 on plasma B cells of naïve mice in comparison to Vac+ mice, although the difference was not significant. However, Vac+ mice had a markedly increased expression of CD93 on bone marrow plasma B cells in comparison to Vac+Sm + mice (p <0.05) (Fig 3E). In addition, plasma B cells of Vac+Sm+PZQ+ expressed higher levels of CD93 than Vac+Sm+ mice and a non-significant difference versus Vac+ mice. These

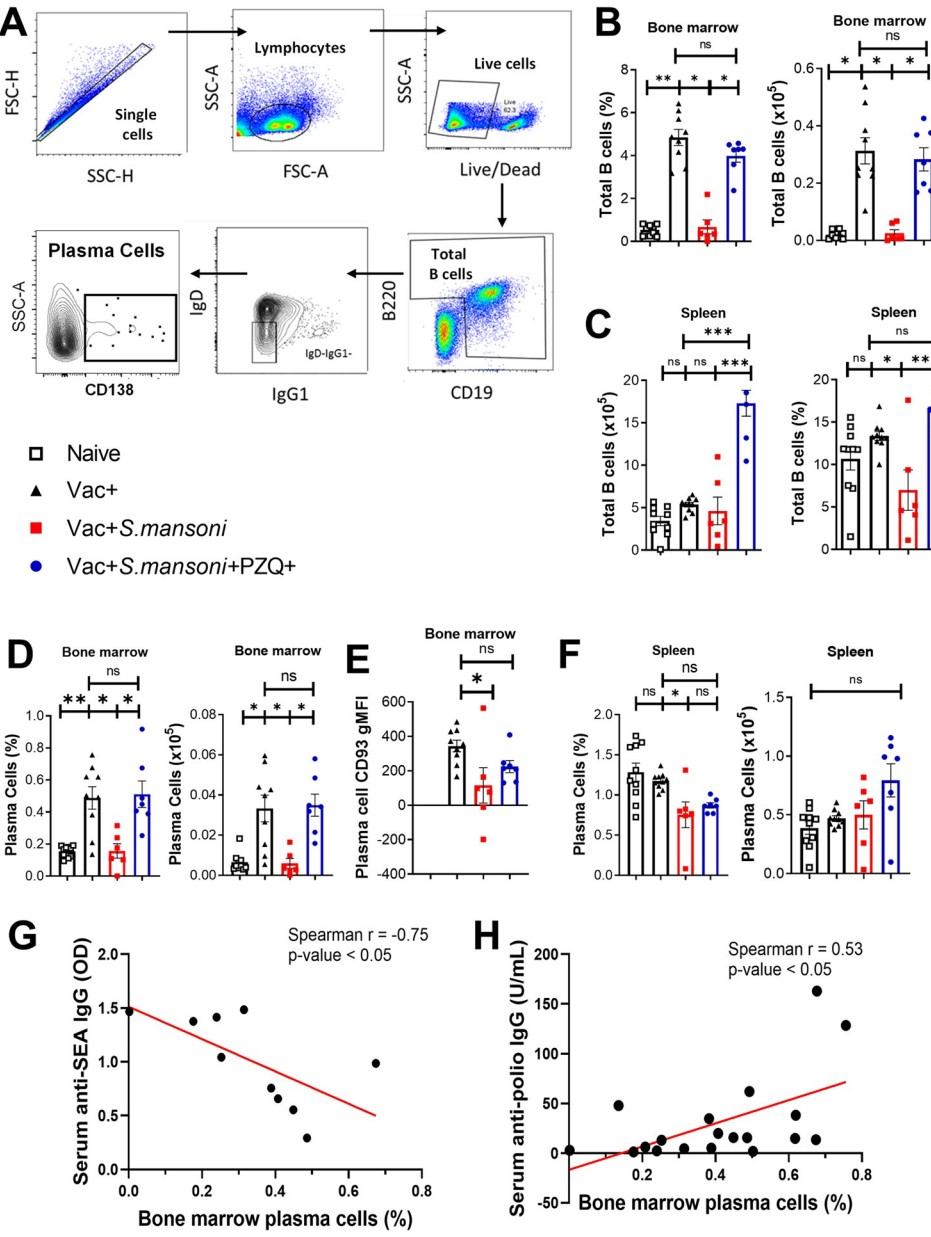

**Fig 3. Total B cell and plasma B cell frequencies and counts at 18 weeks post schistosomiasis infection in the bone marrow and spleens of anti-Poliovirus vaccinated mice.** (A) Representative gating strategy for plasma B cells. (B) Bone marrow total B cell frequencies and total numbers. (C) Spleen B cell frequencies and total numbers. (D) Bone marrow plasma B cells. (E) CD93 gMFI of bone marrow plasma B cells. (F) Spleen plasma B cells. Data are expressed as mean ± S.E.M and representative of 2 independent experiments (n = 6–10 mice per group in each experiment); Data was analysed by the Shapiro-Wilk test followed by either one way ANOVA with Bonferroni's multiple comparisons test or by Kruskal Wallis followed by the Dunn's multiple comparisons test; ns, p> 0.05; * p< 0.05, ** p< 0.001, *** p< 0.0001; Vac, hexavalent (DTPa-hepB-IPV-Hib) vaccine; Sm, *S. mansoni*; PZQ, praziquantel; CD93, cluster of differentiation 93. **Increasing serum anti-SEA IgG titres are associated with reduced bone marrow plasma cell frequencies while increasing anti-polio IgG titres are associated with increasing bone marrow plasma cells.** G. Anti-SEA titres vs bone marrow plasma cell frequencies of schistosomiasis infected Vac+Sm+ and Vac+Sm+PZQ + mice at week 18 p.i. H. Anti-polio IgG titres vs bone marrow plasma cell frequencies of Vac+, Vac+Sm+, and Vac +Sm+PZQ+ mice combined. G-H: Data are representative of 2 independent experiments (n = 4–10 mice per group); Data was analysed by Spearman's correlation.

findings support a bone marrow plasma B cell-survival-associated route for the negative effect of *S. mansoni* infection on the sustainability of anti-polio vaccination-elicited memory responses in mice.

In the spleen, however, there was no significant difference between CD138+ plasma B cell frequencies of Vac+ and naïve mice (Fig 3F). However, Vac+ mice had significantly higher CD138 + plasma B cell frequencies when compared to Vac+Sm+ (p<0.05) (Fig 3F), while no significant differences were observed for absolute numbers. In addition, there was no significant difference between splenic CD138+ plasma B cell frequencies of Vac+ and Vac+Sm+PZQ+ mice and splenic CD138+ plasma B cell frequencies of Vac+Sm+ and Vac+Sm+PZQ+ mice (Fig 3F).

To further assess the tripartite association between infection bone marrow plasma cell frequencies/numbers, serum anti-SEA IgG titres and serum anti-polio IgG titers in our study, we performed associative analyses of these variables (Fig 3G and 3H). We noted that increasing serum anti-SEA IgG titers in *S. mansoni* infected mice negatively associated with bone marrow plasma cell levels (Fig 3G). On the other hand, a positive association between bone marrow plasma cell levels and serum anti-polio IgG titers in was noted in vaccinated mice (Fig 3H).

These observations demonstrated that *S. mansoni* could be associated with reduced antibody-producing cells, the CD138+ plasma B cells, in the bone marrow but not in the spleen of mice. Taken together, we, therefore, hypothesized that the observed reduction of CD93 expression on CD138+ plasma B cells in the bone marrow of *S. mansoni* infected mice might indicate the parasite's driven increased likelihood of bone marrow CD138+ plasma B cell death. We also hypothesized that PZQ treatment and its associated reduction of parasite burden could mitigate the parasite's impact on these antibody producing cells in mice.

## Chronic *S. mansoni* infection is associated with a reduced survival of bone marrow, but not of spleen, plasma B cells of hexavalent (DTPa-hepB-IPV-Hib) vaccinated mice, and anthelminthic chemotherapy PZQ reverses this effect

To further explore the observed reductions of plasma cells in the bone marrow of vaccinated and *S. mansoni* infected mice, and their restoration in the bone marrow of their PZQ-treated counterparts, we assessed the survival of various immune cells important in the generation, maintenance and expansion of plasma cells in the different mice groups. First, the dead cell staining fixable viability dye (FVD) was used to identify and quantify dead plasmablasts and plasma cells by flow cytometry (Fig 4A). The bone marrow of Vac+ mice showed significantly lower frequencies of FVD+CD138+plasma B cells, and lower but non-significant plasmablasts, when compared to naive mice (p<0.01 and p = 0.07, respectively) (Fig 4B and 4C). These results indicated the better survival of plasma cell populations in vaccinated mice when compared to non-vaccinated mice, consistent with a survival advantage of plasma cell lineage following vaccination. Similarly, the bone marrow of Vac+ mice had significantly lower frequencies of FVD+CD138+ dead plasmablasts and plasma cells when compared to the infected Vac+Sm+ mice (p<0. p<0.05 and p<0001, respectively) (Fig 4B and 4C) indicating a parasite-driven provoked death of these cells. Likewise, bone marrow frequencies and absolute numbers of FVD+CD138+ dead plasmablasts and plasma B cells of Vac+ mice were either not different or minimally higher than those of infected and treated Vac+Sm+PZQ+ mice (p>0.05 and p< 0.05, respectively) (Fig 4B and 4C), arguing in favour of a restoration of plasma cells viability following PZQ treatment, especially given the restoration in the Vac+Sm+PZQ + group of levels similar to the Vac+ control group.

Further assessing cellular death in other B cell subsets (S7 Fig), we confirmed total B cell death in the bone marrow (S7B Fig) but moderately so in spleens (S7C–S7F Fig) of

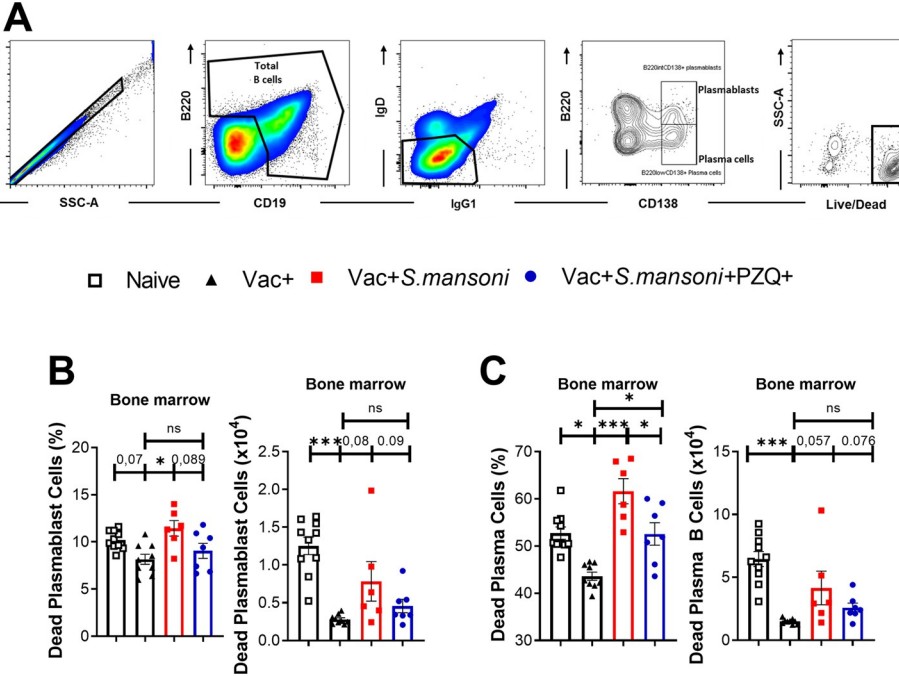

**Fig 4. Cell death in CD138+ plasmablasts and plasma B cells from schistosomiasis-infected and anti-poliovirus vaccinated animals.** (A) Representative gating strategy of dead cells among plasmablasts and plasma cells. (B) Dead plasmablast frequencies and absolute numbers in the bone marrow. (C) Dead plasma cell frequencies and absolute numbers in the spleen. Data are expressed as mean ± S.E.M and representative of 2 independent experiments (n = 6–10 mice per group in each experiment); Data analysed by Shapiro-Wilk test followed by either one way ANOVA with Bonferroni's multiple comparisons test or by Kruskal Wallis followed by the Dunn's multiple comparisons test; ns, p> 0.05; * p< 0.05, ** p< 0.001, *** p< 0.0001; Vac, hexavalent (DTPa-hepB-IPV-Hib) vaccine; Sm, *S. mansoni*; PZQ, praziquantel; SSC, Side scatter; FSC, forward scatter; CD19, cluster of differentiation 19; IgG1, Immunoglobulin G1; IgD, Immunoglobulin D; B220, B cell isoform of 220 kDa; CD138, cluster of differentiation 138.

schistosomiasis diseased mice as this was significantly reversed by PZQ treatment in follicular, IgG1 memory and marginal zone B cells. Altogether, these data suggested that *S. mansoni* infection primarily reduces the survival of antibody-producing plasma B cells in the bone marrow compartment and highlights a corrective function of PZQ-mediated schistosomiasis treatment in countering this effect.

Within the T cell compartment (S8 Fig), a minimal reduction of CD4+ T cells and follicular CD4+ T cells was observed in the spleens of Vac+Sm+ mice whereas PZQ treatment was restorative in Vac+Sm+PZQ+ mice. These observations further supported the decline of humoral responses due to *S. mansoni* infection (S8B and S8C Fig). In parallel, an increase of regulatory Foxp3 expression in splenic CD4+ T cells from Vac+Sm+ mice was noted whereas, upon PZQ treatment in Vac+Sm+PZQ+ mice, the regulatory T cells fell back to lower levels (S8D Fig). These findings were suggestive of an immunomodulatory influence of schistosomiasis in the spleen. However, we failed to note such an effect on Foxp3 expression in bone marrow T cells (S8 Fig), arguing against a direct impact of the Foxp3 compartment within the bone marrow. In tight correlation with these reported dynamics of splenic CD4+ T helper and T follicular helper cells, T cell death (numbers), as measured by FVD+ staining, was minimally represented in the spleens of *S. mansoni* infected mice in comparison to their non-infected counterparts, further arguing against a key role of these T cell subsets in the depletion of B cells during schistosomiasis (S8F and S8G Fig). However, refined assessment of splenic memory T cells in mice (S9 Fig) revealed an elevation of both central memory ($T_{CM}$, S9B Fig) and

effector memory (T$_{EM}$, S9C Fig) CD4+ T cells following vaccination (Vac+ mice) compared to naïve mice (p<0.05, for one-on-one comparison between naïve and Vac+ mice). Mice burdened with schistosomiasis had significantly reduced T$_{CM}$ and T$_{EM}$ in Vac+Sm+ versus Vac + mice while treatment with PZQ was restorative in Vac+Sm+PZQ+ mice (S9B and S9C Fig). These findings indicated a possible remote role of T cell memory in accompanying or supporting the depletion of plasma cells as observed in the bone marrow and spleens of *S. mansoni* infected mice. Intriguingly, CD8+ T$_{CM}$ contrastingly increased (S9D Fig) rather than reduced as observed in their CD4+ counterparts in Vac+Sm+ mice compared to Vac+ mice. Praziquantel appeared to have no impact on CD8+ T$_{CM}$ in Vac+Sm+PZQ+ mice compared to Vac +Sm+ mice. However, *S. mansoni* infection-induced CD8+ T$_{CM}$ remained elevated despite treatment in Vac+Sm+PZQ+mice compared to Vac+ mice (S9D Fig). Conversely, CD8+ Tem frequencies were elevated in Vac+ mice compared to naïve mice but reduced in Vac+Sm + mice versus Vac+ mice (S9E Fig). Treating infected mice with PZQ counteracted parasite's influence to restore CD8+ Tem cell frequencies in Vac+Sm+PZQ+ versus Vac+Sm+ mice. Together, the findings suggest that schistosomiasis alters the T cell compartment of vaccinated mice by depleting follicular T helper cells and memory T cells, and PZQ treatment restores these cells.

## Schistosomiasis-driven death of plasma cells and plasmablasts in the bone marrow is not mediated by apoptosis

Further assessment of the mechanism of cell death in bone marrow plasma cells was performed by dual staining of the cells with FVD live/dead staining and Annexin V staining to evaluate apoptosis (Fig 5A). We failed to note a disproportionate increase of apoptotic plasma cells (Fig 5B–5D) or apoptotic plasmablasts (Fig 5E–5G) in *S. mansoni* infected Vac+ Sm + mice, arguing against the central contribution of apoptosis in the observed schistosomiasis-associated death of bone marrow plasma cells. However, cell death due to membrane-damage, i.e., Annexin V+FVD+ cells, showed a pattern consistent with the differential reduction of plasma cells and plasmablasts in schistosomiasis-diseased Vac+Sm+ mice with an equally recapitulative restoration by death rate reduction in Vac+Sm+PZQ+ mice. Taken together, a non-apoptotic process appears to be at play here to foster plasmablasts and plasma cell death in the bone marrow of mice following schistosomiasis.

## Discussion

Long-term immunity has been demonstrated for poliovirus vaccines [28,29]. Our observations now suggest a negative association between *S. mansoni* infection and the ability of the host to maintain serological memory against poliovirus in vaccinated hosts. This observation is consistent with our recent report of *S. mansoni* infection being associated with reduced responses to the measles virus vaccine in school-aged children from Bokito, Cameroon [8,13]. Additionally, weakened immune responses in *S. mansoni* infected children have been reported for the measles virus vaccine in Uganda [9], and the hepatitis B virus vaccine in Uganda [10] and Egypt [11]. The similarity of these observations for different viral vaccines now suggests a general impairment of long-term serological memory mounted by viral vaccines in *S. mansoni* infected children living in endemic areas. Specifically in the present study, our findings revealed a heightened impact of *S. mansoni* infection with increasing age as we noted a more pronounced effect of *S. mansoni* associated reductions of vaccine-elicited memory responses in older children from five schools combined.

Interestingly, we noted in our study a high prevalence of *S. mansoni* infection (>54%) in children from Y1 in the year 2016, which was considerably reduced to less than 24.2% in the

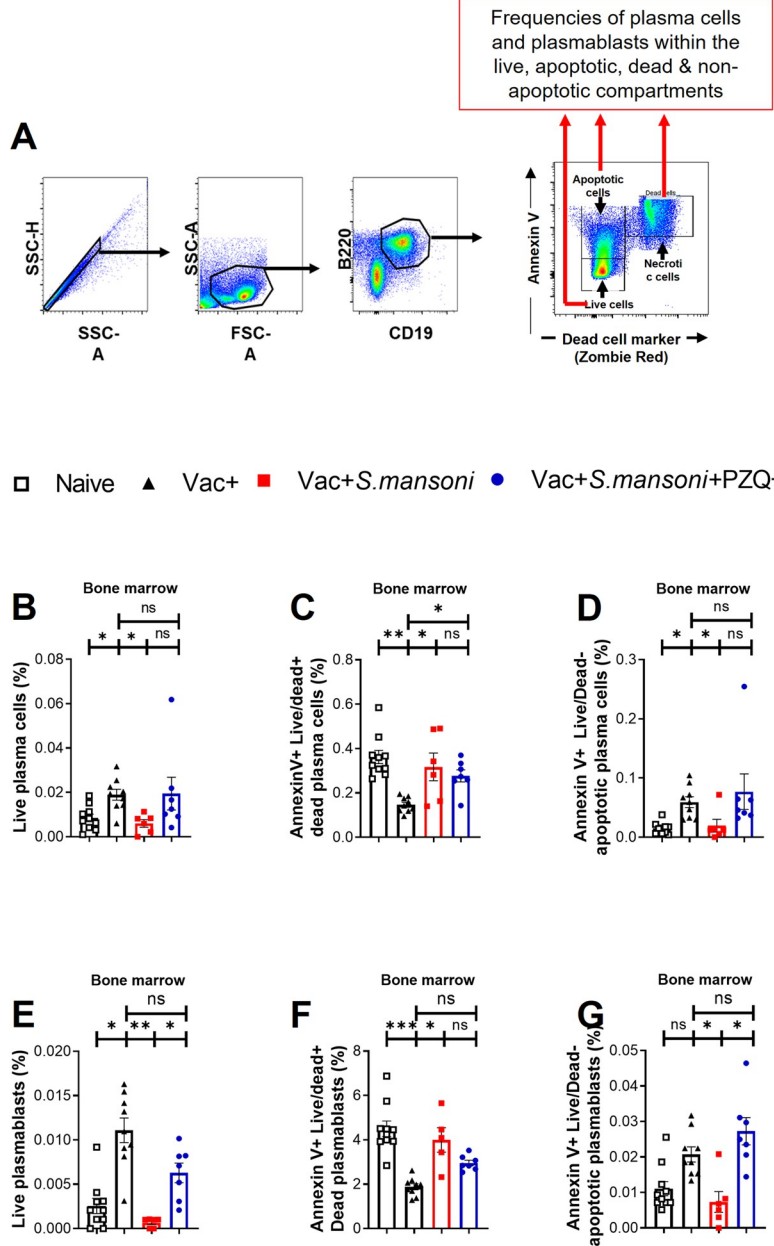

**Fig 5. Schistosomiasis-driven death of plasma cells and plasmablasts in the bone marrow is not mediated by apoptosis.** (A) Representative gating strategy, illustrative, of apoptotic cells (Annexin V+ Zombie Red-), necrotic cells (Annexin V+ Zombie Red+) and live cells (Annexin V- Zombie Red-) that were each sub-gated for frequencies of plasma blasts or plasma cells at week 18 post-infection. Frequencies of plasma cells (CD138+B220-) within the liver (B.), dead (C.) and apoptotic (D.) IgD-IgG1- B cells of the bone marrows. Frequencies of plasmablasts (CD138+B220+) within the live (E.), dead (F.) and apoptotic (G.) IgD-IgG1- B cells of the bone marrows. Data are expressed as mean ± S.E.M and representative of 2 independent experiments (n = 6–10 mice per group in each experiment); The Vac+ group is used throughout as reference for statistical comparison. Data was analysed by two-way ANOVA followed by FDR corrected multiple comparisons against the Vac+Sm+ group; ns, p> 0.05; * p< 0.05, ** p< 0.001, *** p< 0.0001; PZQ, praziquantel.

year 2018. This reduction in the prevalence of *S. mansoni* could be attributed in part to the effectiveness and efficacy of annual PZQ treatments of these children under the MDA campaign in Cameroon, despite the permanent risk of reinfection [8]. The success of MDA in

reducing schistosomiasis prevalence in school children has recently been confirmed for 44 countries in sub-Saharan Africa [30].

We similarly observed that treatment with PZQ could partially reverse anti-viral vaccine specific serological memory in schistosomiasis-diseased children. This was also the case in a recent study in Uganda which demonstrated that PZQ treatment improved responses to the measles catch-up vaccine in *S. mansoni* infected preschool children [9]. This could either denote an intrinsic repair potential of PZQ, irrespective of the infection status and/or a potential linkage to its antiparasitic effect. Our subsequent observation of the restoration of anti-polio vaccine serological responses in children that were confirmed cured of schistosomiasis argues strongly in favour of PZQ's potential ability to reverse serological vaccine memory suppression mediated by killing adult schistosome parasites, an observation further supported by existing reports showing that progressive chronic schistosomiasis is associated with host immunosuppression [31].

We demonstrated that the long-term maintenance of IPV-specific serological memory in DTPa-hepB-IPV-Hib vaccinated mice was consistent with the previous report of long-term maintenance of poliovirus specific serological memory in mice [21]. Utilizing a well-established mouse model to mimic natural human chronic *S. mansoni* infections [32], we showed that infection with chronic *S. mansoni* post vaccination lowers the anti-polio specific serological memory responses over time. Similarly, other studies have demonstrated that *S. mansoni* suppresses mouse immune responses to the diphtheria toxoid [33], the bacille Calmette-Guérin (BCG) vaccine [34] and HIV specific vaccines [35]. In another study, an association between *S. mansoni* infection and reductions of serologic IgG antibodies mounted by the Human Papillomavirus vaccine in non-human primates was described [36]. Highlighting the immunosuppressive nature of different *Schistosoma* spp. in a separate study, chronic *S. japonicum* infection impaired immune responses against the hepatitis B virus vaccine in mice [37]. These observations strengthen the odds that schistosomiasis impairs its host's ability to maintain long-term vaccine memory immunity.

Serological memory immunity is maintained by the continual supply of antibodies produced by plasma cells [23,26,38]. In our study, CD138+ plasma B cells were substantially reduced in the bone marrow and moderately in the spleens of chronic *S. mansoni* infected mice. These observations suggest that the negative impact of *S. mansoni* could be rooted in the bone marrow in addition to satellite defects in the secondary lymphoid organ [13]. Noteworthy, the *S. mansoni* associated plasma cell reduction was well pronounced in bone marrow but poorly in the spleen, thus suggesting the concentrated impact of *S. mansoni* on bone marrow plasma cells through mechanisms that remain unclear. As the bone marrow is a niche for the long-term survival of antibody-producing plasma cells, we reasoned that *S. mansoni* could reduce the survival of these cells. Confirming our hypothesis, CD93, a specific marker of bone marrow plasma cell survival, and not defective migration [23,27], was expressed at markedly lower levels in bone marrow plasma cells of *S. mansoni* infected mice when compared to non-infected counterparts.

Logically, one would have therefore expected the reduced viability of plasma cells expressing lower levels of CD93 in our settings, consistent with compelling reports [27,39]. Measuring the survival of plasma cells in our study, we noted that higher frequencies of the bone marrow CD138+ plasma B cells in chronic *S. mansoni* infected mice were stained with the amine dye for dying or dead cell populations. As such, we consequently concluded that the reduced expression of surface CD93 on plasma cells during *S. mansoni* infection robustly translated into increased death or reduced survival of these cells in mice. A rough parallel could be drawn from human studies where clinical evidence has shown the downregulation of B cell proliferation in *S. haematobium* infected children [40]. Several routes for explanation are envisaged

here to explain the loss of plasma B cells in the bone marrow of patently infected *S. mansoni*-diseased mice. A possibility could be the hijacking of the plasma cell generation/differentiation pipeline during schistosomiasis to foster the expansion/accumulation of another B cell subset. We do, however, find great opposition to this possibility in the assessment of spleen and bone marrow levels of other B cell subsets (marginal, memory and follicular) where no indication of accumulation of any of these B cell subsets was apparent as a result of schistosomiasis. Another explanation could have been an erroneous accumulation of plasma cells in the spleen rather than migrating to the bone marrow to constitute long-lived and powerful antibody-producing cells. However, our results on the levels of plasma cells in the spleen of schistosomiasis-diseased mice do not support an accumulation of these cells in the spleen, arguing against the likelihood of the depletion of plasma cells in the bone marrow being a result of an upstream impaired migration from the spleen. Nevertheless, the observation that splenic T cells, particularly in the Foxp3+ and memory T cell compartment, are altered during schistosomiasis in a way that could favour plasma B cell depletion in the bone marrow do raise the clear possibility that plasma cell depletion might not be sufficient to fully explain the observed reduction of antibody production caused by schistosomiasis. Nonetheless, further investigations on the scope of alteration of schistosomiasis on humoral memory response within the T cell compartment are certainly required despite non-opposing indications on the presently robust observation of a preferential depletion of bone marrow plasmablasts and plasma cells as strong correlates of reduced production of vaccine-elicited antibodies in our setting.

Standing out from our analyses was the possibility of diminished survival and heightened cellular death as demonstrated by dual live/dead and Annexin V staining. Our investigation using this cell death staining combo in bone marrow plasma cells in vaccinated and *S. mansoni* infected mice did not reveal a preferential accumulation of apoptotic plasma cells but rather that of live/dead staining positive dead cells within the plasma cell compartment. These data argue against apoptosis as the central driving force of plasma cell depletion by schistosomiasis. In fact, besides apoptosis, several means of eukaryotic cell death ranging from necrosis/necroptosis, autophagy, pyroptosis, or ferroptosis [41] could be at play here. However, given that markers of cell death can only be internalized as a result of damaged cell membranes and that cellular inflammation was not apparent in the bone marrow of schistosomiasis-diseased mice (no increase in total B cell and total T cell numbers), compelling support is lent to the implication of a membrane-damaging and poorly inflammatory cell death process such as necroptosis or ferroptosis. Clearly, this needs to be carefully investigated in follow-up studies to conclusively understand the mechanistic bases of schistosomiasis-associated plasma cell death in the bone marrow. As of now, nevertheless, our data suggest that the *Schistosoma* spp. parasites could lower the survival of plasmablasts and that of CD138+ plasma B cells in the bone marrow of their hosts.

More pathophysiologically, the discriminative influence on bone marrow rather than splenic plasma cells is intriguing and could well be rooted in the heightened physiological toll chronic infections such as schistosomiasis might have on the bone marrow hematopoietic machinery. Whether chronic schistosomiasis deleteriously solicits the bone marrow hematopoietic machinery as the infection progresses [42] and as such interferes with the maintenance of essential survival cues within the bone marrow for plasma cells such as CD93 expression is to be addressed. Certainly, further experimental evaluations are required to decisively elucidate the mechanism behind these deductions and to validate how far upstream and wide around the B cell activation ladder is the host impaired by *S. mansoni* infection.

Overall, as yet, our mice data suggest that the chronic *S. mansoni* driven reduced survival or increased death of CD138+ plasma B cells in the bone marrow associates with impaired maintenance of vaccine specific serological memories. What remains puzzling and worthy of future

investigations is the selective impairment of polio-specific responses by schistosomiasis whereas SEA-specific humoral responses do persist and are increased over time, indicating a certain level of antigen-specificity in the schistosomiasis-associated impairment of the memory responses.

Whatever the basis for such a sophisticated immunomodulatory ploy of schistosomes, our observations of the impact of *S. mansoni* on poliovirus vaccination in both children and mice raise an important public health concern, as they indicate that global progress made against viral infections through vaccination is at risk. Clinical data have shown that *S. mansoni* lowers vaccine induced memory immunity for the measles virus vaccine in Cameroon [8] and Uganda [9], and the hepatitis B virus vaccine in Uganda [10]. This trend of *S. mansoni* associated impairment of viral vaccines is more concerning, particularly given the global vaccination efforts to control the current SARS-CoV2 pandemic. The control of immunomodulatory infections of the like of *S. mansoni* infections is therefore critical especially in regions where the parasites are endemic.

Our study further revealed that treatment with PZQ partially restored long-term poliovirus vaccine specific serologic memory responses in *S. mansoni*-infected children and mice. Additionally, PZQ treatment improved bone marrow plasma cell responses in mice. These findings support further the utilization of PZQ in schistosomiasis endemic areas despite previous conflicting evidence on its usefulness in ameliorating vaccine responses. Indeed, PZQ has also been shown to improve mice immune responses to the HIV vaccines [35,43] and the hepatitis B virus vaccine [37]. In Olive baboons, PZQ treatment in the chronic stage of *S. mansoni* infection significantly reversed *S. mansoni* associated reductions of HPV specific IgG antibodies [36]. Together, our present findings point at a beneficial effect of PZQ on vaccine responses in schistosomiasis-diseased animals and children. However, further studies assessing the effect of PZQ on vaccine immunity, for example, the current POPVAC trial studies in Uganda [44], are imperative.

In conclusion, the present study showed that *S. mansoni* infection could be associated with impaired vaccine responses in both children and mice, by lowering the survival of CD138 + plasma B cells as shown in mice and ultimately suppressing the maintenance of serological memory. This study further demonstrated that PZQ treatment could partially restore vaccine induced humoral memory responses in *S. mansoni* infected children and cellular and humoral immune responses in mice.

As such, given the remarkable safety profile of PZQ [45], accumulating evidence on the added benefits of repeated MDA with PZQ [46], and the present findings, we support the WHO consideration of intensified PZQ treatments, for populations in schistosomiasis-endemic areas. This strategy could potentially improve vaccination responses in these *S. mansoni* endemic areas and thus could strengthen vaccination benefits in developing countries, particularly regarding the current vaccination efforts against SARS-COV 2 viruses causing the deadly COVID-19 pandemic.

## Material and methods

### Ethics statement

Ethical approval for human studies was granted by the Cameroon National Ethics committee for Human Health Research (Approval no. 2018/02/976/CE/CNERSH/SP). Authorization to conduct the study was also granted by the Ministries of Basic Education (Approval no. 3/105/L/MINEDUB/DREB-C/SDAG/IMS) and Public Health of Cameroon (Approval no. 030-571/L/MINSANTE/SG/DROS/CRC/NPN/TMC/ 631–12.18). The study was conducted with authorization from local, regional and school authorities. Children and parents'/legal guardians were informed of the objectives and the methodology of the study with assistance from

schoolteachers. Participation was voluntary and all children and parents/legal guardians gave informed written consent prior to the study. All collected data were de-identified before downstream analyses to ensure the confidentiality of the participating school children. All school children from the enrolled schools, participant or not, were treated with PZQ as recommended by the Cameroon National program for the control of Schistosomiasis and Soil transmitted Helminthiasis regardless of their parasitological status.

Authorization to conduct experiments on animals was granted by the University of Cape Town Animal Ethics Committee (AER Protocol no. 018/029). All animal experiments were performed to minimize animal suffering in accordance with the guidelines of the Animal Research Ethics committee of the Faculty of Health Science of the University of Cape Town, the South African Veterinary Council (SAVC) and the South African National Standard (SANS 10386:2008).

## Human observational study design, population, and data collection

A cross-sectional study was conducted in five public schools in vicinity to *S. mansoni* infested rivers in 5 different villages namely Yoro 1, Yoro 2, Bongando, Ediolomo and Kedia in Bokito, a rural town located about 100 km from Yaoundé, the capital of Cameroon. This study area has been previously described [47].

The study was conducted in two sub-phases. In the first explorative sub-phase of the study, school children were recruited from Yoro 1 in the year 2016. In the second sub-phase, a follow up of the study with a larger sample size from five different public schools, was carried out in the year 2018. A total of 189 children were recruited from the following schools after exclusion of those infected with malaria, hepatitis B/C and other geohelminths: Yoro 1, Yoro 2, Bongando, Ediolomo and Kedia (S1 Fig). In both phases of the study, school children with at least 6 months of residency within the endemic area and consenting to participate to the study were recruited. In addition, both studies were conducted at least 6 months after PZQ treatment of all children under the Mass Drug Administration (MDA) program of Cameroon. The MDA campaign is aimed at controlling schistosomiasis in endemic regions of Cameroon and is based on the WHO recommended tablet dose pole strategy. The dose of PZQ per child was based on their body height and administered based on the dose-pole to achieve an optimal dosage of 40 mg/kg [48,49]. In addition, all children contacted in our study were treated with a single dosage of 40 mg/kg PZQ regardless of their schistosomiasis status. A total of 8 children were followed up with complete data in both the 2016 and 2018 studies. Of these, 6 children who were infected with *S. mansoni* in 2016 had cleared the parasite in 2018. Further analyses were conducted to assess the impact of parasite clearance by PZQ on vaccination. The remaining 2 children were not infected in 2016 but were found infected in 2018. Due to their small sample size, no further analyses were conducted for these children.

## Standardized Interview administered data collection

Each child was interviewed by a member of the research staff assisted by their class teacher and parents/legal guardians after informed consent was attained. The interviews were carried out and recorded on questionnaires designed to collect demographic information such as age and gender as well as self-reported and/or parent/legal guardian confirmed general health and vaccination status. These deidentified data are available upon request.

## Stool and urine sample collection and parasitological assays

Each child was given 2 prelabelled 50ml screw-cap vials for collection of fresh morning stool samples collected at 2 separate days with a 5-day gap in-between. The stool samples were evaluated for parasitological analyses using the Kato-Katz technique to identify children infected

with *S. mansoni* as previously described [50,51]. Briefly, 41.7mg of stool was prepared for each sample, 2 stools per child, and assessed by microscopy by 2 independent laboratory technologists for the detection and quantification of *S. mansoni* eggs. Urine samples (10ml) were collected from each child and assessed by filtration method to identify *S. haematobium* eggs as previously described [52].

## Blood sample collection and assays

Whole blood (4 ml) was aseptically drawn from each child by venepuncture into Heparin coated tubes by experienced and authorized phlebotomists. Post-puncture care was provided for each child to minimize risk of infection. Malaria infection status was assessed by thick smears as previously described [53]. Briefly, a spot of well mixed blood was placed on a clean slide. Using the edge of another clean slide, red blood cells were carefully lysed into a thick smear to release the *Plasmodium* parasite. The smear was then air-dried followed by Giemsa staining and analysed using an optical microscope. Plasma was isolated from the collected blood by centrifugation and stored at -80˚C until use. HBV and HCV infection statuses were assessed using the DiaSpot HBsAg and DiaSpot HCV-Ab test kits (DiaSpot, Jakarta, Indonesia), respectively, following the manufacturer's instructions. The quantification of total anti-poliovirus IgGs in plasma samples was carried out by sandwich ELISA using a commercial kit (Human Anti-Poliomyelitis Virus 1–3 IgG ELISA Kit, Alpha Diagnostic International, Tx, USA) following the manufacturer's instructions.

## Animal experimental study

**Mice, vaccines, parasites, and treatment.** Mice on a BALB/c background were used. The mice were maintained under specific pathogen-free animal conditions at the University of Cape Town, in accordance with the guidelines established by the Animal Research Ethics committee of the Faculty of Health Science of the University of Cape Town and the South African Veterinary Council (SAVC). Age and gender matched mice aged 6–8 weeks were injected subcutaneously with a commercial hexavalent vaccine against infectious diseases diphtheria, tetanus, pertussis, hepatitis B, poliovirus myelitis (inactivated) and haemophilus influenzae type b conjugate (DTPa-hepB-IPV-Hib) (Sanofi Pasteur, Lyon, France), or were mock vaccinated, at 2 sites i.e., 150 ul on the neck and 100 ul on the abdomen, at day 0 (primary vaccination) and day 30 (secondary vaccination). At day 5 post-secondary vaccination, vaccinated mice were anesthetized and infected by percutaneous abdomen exposure with a low dose of 35 live *S. mansoni* cercariae for 30 minutes as previously described [54]. Live cercariae were generated from *Biomphalaria glabrata* snails (a gift from Adrian Mountford, York, UK), and NMRI female mice were used to maintain and expand the *S. mansoni* parasite larvae. Between week 9 and week 10 post *S. mansoni* exposure, infected mice only were treated with a single daily oral dose of 400mg/kg of PZQ solution (Merck KGaA, Darmstadt, Germany) or mock treated for 7 days. The PZQ solution was prepared as previously described [55]. Briefly, PZQ was weighted at quantities sufficient for 400 mg/kg of animal body weight and mixed with 10 parts 70% Tween + 30% Ethanol using a magnetic stirrer. Next, 90 parts of distilled sterile water were slowly added and stirring to obtain a homogeneous suspension. Administration of 200 ul of the homogeneous suspensions were administered within 3 h after preparation.

## Serum sampling and quantification of anti-poliovirus specific antibody titres by ELISA in mice

From week 4 post *S. mansoni* infection, 50 ul tail vein blood was collected from all mice at week 4, 6, 8, 13, 15 and finally at week 18, the experimental endpoint when mice were

euthanized. Blood was centrifuged in serum separator tubes (BD Bioscience, San Diego, CA) at 8 000 × g for 10 min at 4˚C to retrieve the serum. The upper aqueous serum phase was aliquoted into tubes and stored at -80˚C until further use.

Anti-poliovirus specific total IgG titres were quantified by sandwich ELISA using a commercial kit (Mouse Anti-Poliovirus Virus 1–3 IgG ELISA Kit, Alpha Diagnostic International, Tx, USA) according to the manufacturer's instructions.

## Sampling of the bone marrow and spleen, single cell isolation and flow cytometric analyses

Following euthanasia at week 18 p.i, the rear left femur bone and spleen were collected for each mouse. Single cell suspensions of bone marrow and spleen cells were prepared and immediately used for downstream flow cytometry analyses.

Single cell suspensions from the bone marrow and spleen were stained for surface markers with antibody mixes of the following antibodies: CD19 (APC-Cy7), IgD (FIT-C), CD23 (PE-CY7), CD21/35 (APC) IgG1 (BD Horizon V450), CD138 (BD Horizon APC-R700) and B220 (BD Horizon V500) from BD Pharmingen (San Diego, CA, USA), and CD93 (PerCP-Cy5.5) and Zombie Red Fixable Viability Dye (PE-TR) from Biolegend (San Diego, CA, USA). The FVD was diluted in 1x cold PBS and cells were stained at room temperature. All other antibodies were diluted in FACS buffer (1X PBS with 1% BSA and 0.1% NaN3) with 2% inactivated Rat serum and 2% α-FcγII/III (clone 2.4G2) added to avoid nonspecific binding. The antibody cocktail solutions were used to stain cell suspensions ($1x10^6$) for 30 min on ice. Stained cells were then washed with FACS buffer before resuspension. Acquisition was conducted using BD LSR Fortessa (BD Immunocytometry system, San Jose, CA, USA), and data analysis was performed with FlowJo software (Treestar, Ashland, OR, US).

## Sampling of the liver and gut, and *S. mansoni* egg burden determination

The liver and gut samples were excised and collected from each mouse at week 18 post *S. mansoni* infection. The organs were weighed and *S. mansoni* eggs were counted after tissue digestion using 4% $KOH_{aq}$ for 18 h, as previously described [32]. The intensity of *S. mansoni* infection intensity was expressed as the number of eggs detected per gram of faeces (EPG).

## Statistics

Statistical analyses were conducted using GraphPad Prism 6.0 software (http://www.prism-software.com). Data distribution was assessed by the Shapiro-Wilk test. The Chi-square test was used to assess differences between gender distribution of school children in relation to their *S. mansoni* infection status. All other statistical evaluations were performed using one of the following tests depending on data distribution: unpaired Student's t-test for equal or unequal variances or Mann-Whitney test for comparison between 2 groups, or Kruskal-Wallis test followed by the Dunn's multiple comparisons, or ANOVA followed by Bonferroni's multiple comparisons or FDR corrected multiple comparisons using the two-stage linear step-up procedure of Benjamini, Krieger and Yekutieli [56]. A p-value threshold of < 0.05 was considered as statistically significant.

## Supporting information

**S1 Fig. Flowchart showing the recruitment and assessment of children from 5 public schools in Bokito, a schistosomiasis endemic rural region of Cameroon.** (A) A total of 336 consenting/assenting children, with consent from legal guardians, were enrolled from all

schools combined. All children completed questionnaires and had 2 stool samples collected on two separate days for *S. mansoni egg* detection by the Kato-Katz technique. Plasma was collected only for children without missing data nor infections such as other helminths, malaria, and hepatitis B and/or C. A battery of exams and diagnostic tests were performed on consenting participants and only samples from patients with no missing data for stool examination by Kato Katz, rapid diagnostic testing of Hepatitis B and C viruses and microscopical screening of malaria parasite in blood smears were further used in the present study, as previously described. Finally, 189 children had their plasma samples analysed for anti-poliovirus IgG titres. (B) Samples of children with complete questionnaire data showing complete annual PZQ treatment, and complete anti-poliovirus IgG titre analyses from two studies conducted three years apart were selected. Initial data was collected in the first study conducted in the year 2016 followed by data collection for the same children in a second study in the year 2018. A total of 8 children were identified. These children were treated for schistosomiasis with a dose 40mg/kg of PZQ once in March in the years 2016, 2017 and 2018 under the National Program for the Control of Schistosomiasis and Soil transmitted Helminthiasis of the Ministry of Public Health in Cameroon. Additionally, in 2016, all study participants were treated with a PZQ dose despite infection status, thus totalling 4 doses during the 3-year study period. (TIF)

**S2 Fig. Establishment of a mouse model of anti-poliovirus vaccination.** (A) Experimental design: BALB/c mice (6 to 8 weeks old and acclimatized for 1 week before procedure) were mock vaccinated (naive) or injected subcutaneously with a commercialized hexavalent vaccine, hexavalent (DTPa-hepB-IPV-Hib) vaccine, at 2 sites i.e., 150ul on the neck and 150ul on the abdomen at day 0 and day 30. Blood was collected from tail vein (approximately 50ul) every two weeks until day 142 (week 16 after second vaccine dose). The animals were sacrificed at week 16, terminally bled and their spleens were collected. (B) Serum isolated from blood samples was probed by ELISA for anti-Polio virus IgG titres over time. C. Gating strategy for class switched IgG1 memory B cell. D. IgG1 memory B cells in vaccinated mice in comparison to naïve mice. Data are expressed as mean ± S.E.M and representative of 2 independent experiments (n = 13–14 mice per group in each experiment); Data was analysed by two-way ANOVA followed by FDR corrected multiple comparisons; ns, $p > 0.05$; $^{*}$ $p < 0.05$, $^{**}$ $p < 0.001$, $^{***}$ $p < 0.0001$; Vac, hexavalent (DTPa-hepB-IPV-Hib) vaccine; p.v, post-vaccination. (TIF)

**S3 Fig. Experimental design to assess the influence of schistosomiasis infection on the responsiveness of anti-poliovirus vaccinated mice.** BALB/c mice (6–8 weeks old) were separated into 4 experimental groups. (A) was injected with phosphate buffered saline (PBS). (B-D) Mice were injected subcutaneously with a commercialized hexavalent vaccine, hexavalent (DTPa-hepB-IPV-Hib) vaccine, at 2 sites i.e., 150ul on the neck and 100ul on the abdomen, at day 0 and day 30. Five days after the second vaccine dose (day 35), Naïve (A) and IPV-Vaccinated control mice (B) were mock infected with PBS while groups (C) (vaccinated then *S. mansoni* infected i.e., Vac+Sm+) and (D) (Vaccinated then *S. mansoni* infected then PZQ treated i.e., Vac+Sm+PZQ+) were percutaneously infected with a low dose of *S. mansoni* (35 cercariae) to establish a chronic disease course. At the beginning of week 10 after infection (end of week 9 p.i), (A-C) were treated with PBS while (D) was treated with 400mg/kg PZQ once daily for one week. From week 4 to week 8, and from week 13 to week 18 (experimental endpoint) p.i, the animals were bled every 2 weeks from the tail vein and serum was obtained. Animals were monitored for schistosomiasis disease progression (daily weight over time) from day zero of infection. The animals were euthanised at week 18 p.i and cardiac blood and organs (spleen, bone marrow, liver, and gut) were collected for further analyses. Vac,

hexavalent (DTPa-hepB-IPV-Hib) vaccine; Sm, *S. mansoni*; PZQ, praziquantel; p.i, post *S. mansoni* infection.
(TIF)

**S4 Fig. Increase in *S. mansoni* egg burden and serum SEA IgG titers are associated with reduced serum anti-polio vaccine induced IgG titers in mice.** (A) Anti-polio IgG titres vs anti-SEA IgG titres at week 4 p.i. (B) Anti-polio IgG titres vs anti-SEA IgG titres at week 8 p.i. (C) Anti-polio IgG titres vs anti-SEA IgG titres at week 18 p.i. (D) Anti-polio IgG titres vs liver EPG at week 18 p.i. (E) Anti-polio IgG titres vs gut EPG at week 18 p.i. (F) SEA-specific antibodies in serum at week 4 p.i. (G) SEA-specific antibodies in serum at week 8 p.i. (H) SEA-specific antibodies in serum at week 18 p.i. (I) Serum anti-SEA IgG titre kinetics response to schistosomiasis infection at week at week 4, 8 and 18. Data are expressed as mean ± S.E.M and representative of 2 independent experiments (n = 6–10 mice per group in each experiment); The Vac+ group is used throughout as reference for statistical comparison. Data was analysed by two-way ANOVA followed by FDR corrected multiple comparisons against the Vac+Sm + group; ns, $p > 0.05$; $^*$ $p < 0.05$, $^{**}$ $p < 0.001$, $^{***}$ $p < 0.0001$; PZQ, praziquantel; p.i, post schistosomiasis infection.
(TIF)

**S5 Fig. Chronic schistosomiasis depletes bone marrow plasma cells early in mice.** BALB/c mice (6–8 weeks old) were percutaneously infected with a low dose of *S. mansoni* (35 cercariae) to establish a chronic disease course. Animals were sacrificed at week 4 or week 10 post-infection and B cell populations were quantified in bone marrow and spleen. For week 4 post-infection, (A) Total Bone marrow B cell frequencies and numbers; (B) Total Bone marrow plasmablast frequencies and numbers; (C) Total Bone marrow plasma B cell frequencies and numbers; For week 10 post-infection, (D) Total Bone marrow B cell frequencies and numbers; (E) Total Bone marrow plasmablast frequencies and numbers; (F) Total Bone marrow plasma B cell frequencies and numbers; (G) Liver egg burdens at week 4 and week 10 p.i; (H) Liver and spleen index weights (as a ratio of total body weights) from naïve and infected mice at week 4 vs. week 10 p.i. Data (n = 3–6 mice per group) are expressed as mean ± S.E.M; Data analysed One way ANOVA by followed by the Bonferroni's multiple comparisons test or by Kruskal wallis followed by the Dunn's multiple comparisons test; ns, $p > 0.05$; $^*$ $p < 0.05$, $^{**}$ $p < 0.001$, $^{***}$ $p < 0.0001$; Sm, *S. mansoni*; PZQ, praziquantel; p.i, post *S. mansoni* infection.
(TIF)

**S6 Fig. Frequencies and numbers of Total B cells, IgG1 memory B cells, Follicular B cells and Marginal zone B cells.** (A) Representative flow cytometry analysis of total B cells from bone marrow and spleen, and IgG1 memory B cells, follicular B cells and MZ B cells from the spleen. (B) Total B cells in the bone marrow. (C) Total B cells in the spleen. (D) IgG1 memory B cells in the spleen. (E) Follicular B cells in the spleen. (F) MZ B cells in the spleen. Data are expressed as mean ± S.E.M and representative of 2 independent experiments (n = 6–10 mice per group in each experiment); Data was analysed by the Shapiro-Wilk test followed by either one way ANOVA with Bonferroni's multiple comparisons test or by Kruskal Wallis followed by the Dunn's multiple comparisons test; ns, $p > 0.05$; $^*$ $p < 0.05$, $^{**}$ $p < 0.001$, $^{***}$ $p < 0.0001$; Vac, hexavalent (DTPa-hepB-IPV-Hib) vaccine; Sm, *S. mansoni*; PZQ, praziquantel; SSC, Side scatter; FSC, forward scatter; CD19, cluster of differentiation 19; IgG1, Immunoglobulin G1; IgD, Immunoglobulin D; B220, B cell isoform of 220 kDa; CD93, cluster of differentiation 93; CD21/35, cluster of differentiation 21 and 35; CD23, cluster of differentiation 23.
(TIF)

**S7 Fig. Frequencies and numbers of dead Total B cells, IgG1 memory B cells, Follicular B cells and Marginal zone B cells.** (A) Representative flow cytometry analysis of dead total B cells from bone marrow and spleen, and dead IgG1 memory B cells, follicular B cells and MZ B cells from the spleen. (B) Dead total B cells in the bone marrow. (C) Dead total B cells in the spleen. (D) Dead IgG1 memory B cells in the spleen. (E) Dead follicular B cells in the spleen. (F) Dead MZ B cells in the spleen. Data are expressed as mean ± S.E.M and representative of 2 independent experiments (n = 6–10 mice per group in each experiment); Data was analysed by the Shapiro-Wilk test followed by either one way ANOVA with Bonferroni's multiple comparisons test or by Kruskal Wallis followed by the Dunn's multiple comparisons test; ns, $p > 0.05$; * $p < 0.05$, ** $p < 0.001$, *** $p < 0.0001$; Vac, hexavalent (DTPa-hepB-IPV-Hib) vaccine; Sm, *S. mansoni*; PZQ, praziquantel; SSC, Side scatter; FSC, forward scatter; CD19, cluster of differentiation 19; IgG1, Immunoglobulin G1; IgD, Immunoglobulin D; B220, B cell isoform of 220 kDa; CD93, cluster of differentiation 93; CD21/35, cluster of differentiation 21 and 35; CD23, cluster of differentiation 23.
(TIF)

**S8 Fig. Frequencies and numbers of CD4+ T cells, CD4 follicular T cells and Foxp3+ T cells.** (A) Representative flow cytometry gating strategy for identifying CD3+CD4+ T helper cells and BCL-6+CD3+CD4+follicular T helper cells. (B) CD4+ T helper cells in the spleen. **C.** BCL-6+CD3+CD4+follicular T cells in the spleen. (D) Foxp3 gMFI expression on CD3+CD4 + T cells in the spleen and bone marrow. (E) Representative flow cytometry gating strategy for identifying dead CD3+CD4+ T helper cells and BCL-6+CD3+CD4+follicular T cells in the spleen. (F) Dead splenic CD3+CD4+ T helper cells. (G) Dead splenic BCL-6+CD3+CD4+-follicular T helper cells in the spleen. Data are expressed as mean ± S.E.M and representative of 2 independent experiments (n = 6–10 mice per group in each experiment); Data was analysed by the Shapiro-Wilk test followed by either one way ANOVA with Bonferroni's multiple comparisons test or by Kruskal Wallis followed by the Dunn's multiple comparisons test; ns, $p > 0.05$; * $p < 0.05$, ** $p < 0.001$, *** $p < 0.0001$; Vac, hexavalent (DTPa-hepB-IPV-Hib) vaccine; Sm, *S. mansoni*; PZQ, praziquantel; SSC, Side scatter; CD3, cluster of differentiation 3; CD4, cluster of differentiation 4; CD8, cluster of differentiation 8; BCL-6, IgG1, Immunoglobulin G1; IgD, Immunoglobulin D; B220, B cell isoform of 220 kDa; CD138, cluster of differentiation 138, BCL-6, B-cell lymphoma 6; Foxp3, forkhead box P3.
(TIF)

**S9 Fig. Frequencies and numbers of splenic central memory and effector memory T cells.** (A) Representative flow cytometry gating strategy for identifying central memory, effector memory and naive T cells. (B) CD3+CD62L+CD44+CD4+ central memory T cells. (C). CD3 +CD62L-CD44+CD4+ effector memory T cells. (D) CD3+CD62L+CD44+CD8+ central memory T cells. (E) CD3+CD62L-CD44+CD8+ effector memory T cells. Data are expressed as mean ± S.E.M and representative of 2 independent experiments (n = 6–10 mice per group in each experiment); Data was analysed by the Shapiro-Wilk test followed by either one way ANOVA with Bonferroni's multiple comparisons test or by Kruskal Wallis followed by the Dunn's multiple comparisons test; ns, $p > 0.05$; * $p < 0.05$, ** $p < 0.001$, *** $p < 0.0001$; Vac, hexavalent (DTPa-hepB-IPV-Hib) vaccine; Sm, *S. mansoni*; PZQ, praziquantel; SSC, Side scatter; CD3, cluster of differentiation 3; CD4, cluster of differentiation 4; CD8, cluster of differentiation 8; $T_{CM}$, central memory T cells; $T_{EM}$, effector memory T cells.
(TIF)

**S1 Table. Schistosomiasis prevalence: Distribution of children by sex, age groups and *S. mansoni* infection status.**
(DOCX)

## Acknowledgments

We are very grateful to all children who participated in this survey, their parents/authorized legal guardians, the Directors and teachers of the primary schools surveyed. Special gratification to all authorities who granted us with required authorization. We also acknowledge the National Public Health Laboratory of the Ministry of Public Health in Yaoundé, Cameroon where some preliminary analyses were performed particularly. The authors are also grateful to Merck Global Health Institute, through Dr. Thomas Spangenberg, which helped source Praziquantel from Merck KGaA donation program in Cameroon for the treatment of all schoolchildren of the study. Special thanks are finally due to the National Program for the Control of Schistosomiasis and Soil transmitted Helminthiasis through the Ministry of Public Health and the Ministry of Basic Education of Cameroon for the relentless effort in regularly treating schoolchildren in Cameroon. We are grateful to the UCT Animal Unit staff for the maintenance of mice. We thank Ms Munadia Ansarie and Ms Lizette Fick for their valuable technical assistance with animal breeding and histology, respectively.

## Author Contributions

**Conceptualization:** Justin Komguep Nono.

**Data curation:** Fungai Musaigwa, Severin Donald Kamdem, Thabo Mpotje, Justin Komguep Nono.

**Formal analysis:** Fungai Musaigwa, Severin Donald Kamdem, Thabo Mpotje, Justin Komguep Nono.

**Funding acquisition:** Frank Brombacher, Justin Komguep Nono.

**Investigation:** Fungai Musaigwa, Severin Donald Kamdem, Thabo Mpotje, Paballo Mosala, Nada Abdel Aziz, Justin Komguep Nono.

**Methodology:** Fungai Musaigwa, Severin Donald Kamdem, Thabo Mpotje, Paballo Mosala, Nada Abdel Aziz, Justin Komguep Nono.

**Project administration:** De'Broski R. Herbert, Justin Komguep Nono.

**Resources:** Frank Brombacher, Justin Komguep Nono.

**Supervision:** Severin Donald Kamdem, De'Broski R. Herbert, Justin Komguep Nono.

**Validation:** Fungai Musaigwa, Severin Donald Kamdem, Justin Komguep Nono.

**Writing – original draft:** Fungai Musaigwa, Severin Donald Kamdem, Thabo Mpotje, De'Broski R. Herbert, Frank Brombacher, Justin Komguep Nono.

**Writing – review & editing:** Fungai Musaigwa, Severin Donald Kamdem, Justin Komguep Nono.

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
