## [Decision Letter · Decision Letter 0]

8 Oct 2021

Dear Dr. Nono,

Thank you very much for submitting your manuscript "Schistosomiasis induces plasmablast and plasma cell death in the bone marrow and accelerates the decline of host vaccine responses" for consideration at PLOS Pathogens. As with all papers reviewed by the journal, your manuscript was reviewed by members of the editorial board and by several independent reviewers. In light of the reviews (below this email), we would like to invite the resubmission of a significantly-revised version that takes into account the reviewers' comments.

Both reviewers provided helpful feedback, with Reviewer 1 asking for minor changes, and Reviewer 2 asking for necessary additional experiments to support the authors' conclusions. Based on these, recommendations are to: 1/ Address all minor requests 2/ Address major issues, specifically those involving controls including (i) Schistosome-specific antibodies, (ii) PZQ treatment of uninfected group, (iii) more in-depth parameters to assess cell death (i.e. not just the live/dead stain). Given that there may be logistic caveats to measurements from the clinical specimens, some of these requests could be carried out in the mouse model, although supporting data in the clinical study would be helpful.

We cannot make any decision about publication until we have seen the revised manuscript and your response to the reviewers' comments. Your revised manuscript is also likely to be sent to reviewers for further evaluation.

Sincerely,

Meera Goh Nair

Associate Editor

PLOS Pathogens

James Collins III

Section Editor

PLOS Pathogens

Kasturi Haldar

Editor-in-Chief

PLOS Pathogens

orcid.org/0000-0001-5065-158X

Michael Malim

Editor-in-Chief

PLOS Pathogens

orcid.org/0000-0002-7699-2064

Both reviewers provided helpful feedback, with Reviewer 1 asking for minor changes, and Reviewer 2 asking for necessary additional experiments to support the authors' conclusions. Based on these, recommendations are to: 1/ Address all minor requests 2/ Address major issues, specifically those involving controls including (i) Schistosome-specific antibodies, (ii) PZQ treatment of uninfected group, (iii) more in-depth parameters to assess cell death (i.e. not just the live/dead stain). Given that there may be logistic caveats to measurements from the clinical specimens, some of these requests could be carried out in the mouse model, although supporting data in the clinical study would be helpful.

Reviewer's Responses to Questions

**Part I - Summary**

Reviewer #1: In the manuscript the authors investigated the involvement of schistosoma mansoni infection on host anti-poliovirus vaccine response. They suggest that schistosomiasis may induce plasmablast and plasma cell death which limit blood plasma anti-poliovirus specific IgG antibody titer. However, this study is likely to be make less sense. We all know that poliovirus mainly affects children under the age of 6, but the plasma anti-polio IgG antibody titer showed no difference after schistosoma mansoni infection until the children reach the age of 14 in this study. Most important, there are multiple major flaws in study design, especially some critical control groups were not included, and the data that they present is far from supporting their conclusions.

Reviewer #2: The present study is a very good study which is novel and shows important significance.

The present study examined the impact of chronic schistosomiasis on the sustain-ability of vaccine-induced immunity in both children living in endemic areas and experimental infections in mice. Their results demonstrated that showed that S. mansoni infection could be associated with impaired vaccine responses in both children and mice, by lowering the survival of CD138+ plasma B cells as shown in mice, and ultimately suppressing the maintenance of serological memory. This study further demonstrated that PZQ treatment could restore vaccine induced humoral memory responses in S. mansoni infected children, and cellular and humoral immune responses in mice.

However, there are minor grammatic errors in the manuscript. After authors revise it, I suggest the manucript be accepted for publishment.

**Part II – Major Issues: Key Experiments Required for Acceptance**

Reviewer #1: 1. In researches, diagnostic negative only based on single stool examination (Kato Katz) is not reliable. It principally needs to combine with other methods or strategies to efficiently improve the accuracy of diagnosis, e.g. each candidate needs ELISA plus three/five stools for nine/ten detections. However, it seems likes authors did not follow these principals.

2. Schistosome antigens (SWA and SEA)-specific antibodies must be used as controls to be detected together with anti-poliovirus antibody, which is critical to clarify their hypothesis.

3. A critical control group, Naive+PZQ, was missing in both human and animal experiments. Because PZQ has direct impact on immune system, which may increase B and plasma cells in bone marrow and spleen directly but not indirectly by killing of adult worm. In addition, eggs in host liver and intestine play far important roles in altering host immune responses after infection, while PZQ only efficiently kills adult worm instead of eggs, which further suggests a possibility that PZQ impact on host immune response directly.

4. The immune regulation in schistsomiasis is intensively studied but the contribution of plasmablast and plasma cell death, if there is any, to the decreased antibody level is not certain. For example, lower level of anti-poliovirus antibody may also results from schistosome infection-induced Tregs or infection-induced competition of B cell differentiation, which may recover when treat mice with PZQ.

5. The single antibody level detection data in Figures 1 & 2 are not strong enough to demonstrate the serological memory of a vaccine. More important mediators and comprehensive indicators are needed, such as memory B lymphocytes, plasma cells, bystander T cells, etc.

6. Current results in Fig 1 are not sufficient to support conclusion. For example, Fig 1C missed two critical control groups: 1) anti-poliovirus antibody levels from persons without Sm infection at 2016 and 2018 which would help to know whether the antibody level is still in stage of arising. 2) anti-poliovirus antibody levels from PZQ-treated normal person without Sm infection, which may exclude the direct regulation of PZQ on immune system that have been reported in recent years. If there were a group of PZQ+reinfection will be helpful to support their conclusion.

7. The authors tried to conclude the IPV-specific plasmablast and plasma cell apoptosis during schistosome infection, unfortunately, the authors failed to show the “specific”. More important, they failed to detect the death in any other import immune cell types as controls. In addition, authors only used an method which is impossible to distinguish the types of cell death.

8. The results in the manuscript are simply descriptive, mechanisms about how schistosoma mansoni infection induces plasmablast and plasma cell death is not involved at all.

9. Schistosomes are very special worms that quite different from other helminths. Schistosomes lives in blood vessels, which allows the worm to release excreted secretions directly into the blood. And some of the secretions are toxic to the human cells that may include plasmablast and plasma cell. However, it is most likely that this will not happen when hosts infect with other helminths. So, it is not accurate for the authors to state “to uncover the immunological interactions between the vaccine-elicited responses and helminth infection” in line 92. As a better choice, the patients listed in exclusion criteria (Lines 116-118), especially patients co-infected with geohelminths, could be set as control groups.

Reviewer #2: No.

**Part III – Minor Issues: Editorial and Data Presentation Modifications**

Reviewer #1: 1. Line 142, “impactserological” should be “impact serological”. Line 250, “restoredvaccine-induced” should be “restored vaccine-induced”. Line 267, “cellsin” should be “cells in”.

2. Fig 2. This figure should be moved to supplementary material. What solution or vesicle was used in the mock injection group? “Naïve” should be “Mock”?

3. Fig 3.

1) 3A-3D suggest serious pathogenesis in mice livers and intestines in group of Vac+Sm without PZQ treatment, which may significantly impact on the synthesis of antibodies. Thus, Schistosome antigens (SWA and SEA)-specific antibodies must be used as controls.

2) 3F, the author state that “PZQ treatment significantly restored vaccine-induced IgG antibody titres”, however, the antibody titre of PZQ treatment group is still significantly much lower than that of Vac+ group, so what is the effect of this restoration?

4. Fig 4.

1) A critical control group, Naiver+PZQ, was missing, because PZQ has direct impact on immune system, which may increase B and plasma cells in bone marrow and spleen directly but not indirectly by killing of adult worm. In addition, eggs in host liver and intestine play far important roles in altering host immune responses after infection, while PZQ only efficiently kills adult worm instead of eggs, which further suggests a possibility that PZQ impact on host immune response directly.

2) Two important control groups, Sm and Sm+PZQ, were missing.

3) 4B vs 4C, why vaccination only increased B cells in bone marrow but not in spleen?

4) 4B\\4C\\4D, there is no increase of B and plasma cells either in bone marrow or spleen after Sm infection (actually the exact group was missing), thus, how antibodies comes?

5) 4F, results showed there is no significant CD138+plasma cells were induced in Vac+ group, how antibodies comes?

6) 4F, the percentage and number of the cells in PZQ treatment group are higher than or similar to those of Vac+ group, but the antibody titre in PZQ treatment group is significantly lower than that of Vac+ group (eg. Fig 3E). These results are contradictory.

5. Fig 5.

1) Two important control groups, Sm and Sm+PZQ, were missing.

2) 5B and 5C showed that the mock/naïve treatment resulted in the biggest numbers of dead plasmablast and plasma cells in bone marrow, however Vac, Vac+Sm, and Vac+Sm+PZQ saved the cells from death?

3) Does the death of antibody-producing plasma B cells includes schistosome antigen-specific antibody-producing plasma B cells? What about paralleled levels of schistosome antigen-specific antibodies?

6. Discussion.

1) What type of cell death is of interesting, e.g. apoptosis, necrosis, pyroptosis, or ferroptosis? Does the generation and/or differentiation of cells contribute to the decrease? These issues needs to be discussed.

2) Line 386. Without detection of any other antibodies, current data in the manuscript is not enough to support the conclusion like “……IPV specific serological memory……”. Similarly, all words of “specific” in the manuscript need to be carefully evaluated.

Reviewer #2: There are minor grammatic errors in the manuscript.

PLOS authors have the option to publish the peer review history of their article (what does this mean?). If published, this will include your full peer review and any attached files.

Reviewer #1: No

Reviewer #2: No
---

## [Decision Letter · Decision Letter 1]

21 Jan 2022

Dear Dr. Nono,

Thank you very much for submitting your manuscript "Schistosomiasis induces plasmablast and plasma cell death in the bone marrow and accelerates the decline of host vaccine responses" for consideration at PLOS Pathogens. As with all papers reviewed by the journal, your manuscript was reviewed by members of the editorial board and by several independent reviewers. The reviewers appreciated the attention to an important topic. Based on the reviews, we are likely to accept this manuscript for publication, providing that you modify the manuscript according to the review recommendations.

Please address the minor comments from the reviewer.

Sincerely,

Meera Goh Nair

Associate Editor

PLOS Pathogens

James Collins III

Section Editor

PLOS Pathogens

Kasturi Haldar

Editor-in-Chief

PLOS Pathogens

orcid.org/0000-0001-5065-158X

Michael Malim

Editor-in-Chief

PLOS Pathogens

orcid.org/0000-0002-7699-2064

Please address the minor comments from the reviewer.

Reviewer Comments (if any, and for reference):

Reviewer's Responses to Questions

**Part I - Summary**

Reviewer #1: The authors have performed additional experiments and amended the manuscript to sufficiently address the majority of my concerns.

**Part II – Major Issues: Key Experiments Required for Acceptance**

Reviewer #1: No

**Part III – Minor Issues: Editorial and Data Presentation Modifications**

Reviewer #1: 1. The title of the manuscript that "Schistosomiasis induces plasmablast and plasma cell death in the bone marrow and accelerates the decline of host vaccine responses". It is inappropriate and should be changed to: “Schistosoma mansoni infection induces plasmablast and plasma cell death in the bone marrow and accelerates the decline of host vaccine responses”.

2. Sample size in each of the experiments should be explicitly mentioned throughout the results and should be included (n=) in the figure legends as much as possible. This is crucial to assess the robustness of the findings and put in perspective statistical significance.

3. The discussion seems messy and it should be even more concise.

4. Also, there are still a number of language and typo errors throughout. I would still recommend revise language throughout the manuscript.

e.g.,

line 99: a S. mansoni

line 101: schistosomiasis infection

line 249: at week 4, 8, and 18 p.i. ……

line 793: using the the two-stage

etc.

PLOS authors have the option to publish the peer review history of their article (what does this mean?). If published, this will include your full peer review and any attached files.

Reviewer #1: No

Figure Files:

Data Requirements:

Reproducibility:

References:

---

## [Editor Report · Decision Letter 2]

1 Feb 2022

Dear Dr. Nono,

We are pleased to inform you that your manuscript 'Schistosoma mansoni infection induces plasmablast and plasma cell death in the bone marrow and accelerates the decline of host vaccine responses' has been provisionally accepted for publication in PLOS Pathogens.

Best regards,

Meera Goh Nair

Associate Editor

PLOS Pathogens

James Collins III

Section Editor

PLOS Pathogens

Kasturi Haldar

Editor-in-Chief

PLOS Pathogens

orcid.org/0000-0001-5065-158X

Michael Malim

Editor-in-Chief

PLOS Pathogens

orcid.org/0000-0002-7699-2064

Thank you for addressing the minor revision requests, which has improved the accuracy and readability of the manuscript.
---

## [Editor Report · Acceptance letter]

10 Feb 2022

Dear Dr. Nono,

We are delighted to inform you that your manuscript, "Schistosoma mansoni infection induces plasmablast and plasma cell death in the bone marrow and accelerates the decline of host vaccine responses," has been formally accepted for publication in PLOS Pathogens.

Best regards,

Kasturi Haldar

Editor-in-Chief

PLOS Pathogens

orcid.org/0000-0001-5065-158X

Michael Malim

Editor-in-Chief

PLOS Pathogens

orcid.org/0000-0002-7699-2064